# A Review of Differential Plant Responses to Drought, Heat, and Combined Drought + Heat Stress

**DOI:** 10.3390/cimb47120975

**Published:** 2025-11-24

**Authors:** Nankai Li, Zhi Geng, Xiaodong Huang, Shunqi Huang, Lulu Song, Ruirui Chen, Ziping Chen, Liji Du, Congshan Xu

**Affiliations:** 1Anhui Science and Technology Achievement Transformation Promotion Center, Anhui Provincial Institute of Science and Technology, Hefei 230031, China; zhenzhidashu66@163.com (N.L.); huangxiaodong1996@163.com (X.H.); huangshunqi2016@163.com (S.H.); 18326652324@163.com (L.S.); 13856920744@163.com (R.C.); zpchenhf@163.com (Z.C.); leonardo1317@163.com (L.D.); 2College of Agronomy, Nanjing Agricultural University, Nanjing 210014, China; 2022201102@stu.njau.edu.cn; 3Sanya Institute of Nanjing Agriculture, Jiangsu Collaborative Innovation Center for Modern Crop Production, Key Laboratory of Crop Physiology Ecology and Production Management, Nanjing Agricultural University, Nanjing 210095, China; 4Anhui Province Institute of Product Quality Supervision & Inspection, Hefei 230051, China

**Keywords:** drought stress, heat stress, combined stress, physiological response, biochemical regulation, molecular mechanisms, signal transduction

## Abstract

Global warming increases the frequency with which drought and heat stress occur simultaneously, especially in semi-arid regions. Such combined stress imposes a non-additive and more severe impact on plant growth, yield, and quality than either stress alone. Here, we integrate recent physiological, biochemical, and multi-omics studies to compare individual and combined stress responses and to dissect the underlying signal transduction networks. We show that drought-dominated phases rapidly elevate ABA concentrations and activate SnRK2–AREB cascades, whereas heat pulses trigger jasmonic acid and ethylene signals that antagonize ABA-driven stomatal closure. Under combined stress, these hormonal modules converge on a “competitive TF marketplace”, where ABA, JA, and GA cis-elements co-regulate invertase–sugar checkpoints, heat shock factor/ROS oscillators, and chromatin-remodeling events that determine reproductive fate. Recent advances using multi-omics approaches and systems biology have further elucidated these complex networks. These insights will inform future breeding strategies aiming to develop stress-tolerant crops. We highlight emerging tools—weighted gene co-expression networks, kinetic multi-omics, and cis-regulatory CRISPR editing—that can exploit these signaling hubs for breeding crops with improved combined stress tolerance.

## 1. Introduction

Owing to global warming and the increasing frequency of extreme weather events, drought and heat have become two major factors limiting plant productivity and sustainable agriculture [1,2]. In many regions—especially semi-arid areas—drought and heat often occur simultaneously, leading to severe yield losses [3,4]. While individual stress mechanisms have been extensively studied, recent research indicates that responses to combined drought + heat stress invoke additional regulatory mechanisms that extend beyond a simple additive effect (Figure 1) [5,6,7]. Notably, emerging data on signal transduction pathways, such as the MAPK cascade, calcium signaling, and reactive oxygen species (ROS) as secondary messengers, have provided deeper insights into how plants perceive and integrate these stresses [7,8,9]. Building on these advances, our previous studies provided an in-depth physiological analysis of plant responses to both drought and heat stress, elucidating key regulatory nodes that link membrane sensing to whole-plant performance (Figure 2A,B) [9]. This finding is corroborated by several other studies that have also identified similar regulatory mechanisms in different plant species, highlighting the conserved nature of these pathways across diverse taxa [3,4,5,6,7].

Both drought and heat stress induce rapid physiological, biochemical, and molecular changes in plants. Drought stress typically reduces stomatal conductance, lowers transpiration rates, and decreases photosynthetic activity, as plants close their stomata to conserve water [1,5,6,7]. In contrast, heat stress primarily impairs photosynthesis and disrupts cellular homeostasis by increasing leaf temperatures and damaging the photosynthetic apparatus [2,10]. While drought induces stomatal closure, heat stress transiently promotes stomatal opening, leading to non-linear changes in water-use efficiency [2,11]. Combined drought + heat stress, which frequently occurs under field conditions, imposes compounded and complex challenges on plants [3,4,8]. Under these conditions, the interplay between drought-induced stomatal closure and heat-induced stomatal opening creates conflicting signals, leading to non-linear physiological responses [1,12]. Consistent with the temperature overshoot that we previously observed under combined stress, drought + heat stress causes greater physiological disruption than either stress alone [1,2,9,13]; however, this synergy cannot be explained by a simple additive mechanism. This review aims to further consolidate and compare the physiological, biochemical, and molecular responses of plants under drought, heat, and combined stress conditions, with a particular focus on the signal transduction networks that underpin these responses, and to discuss implications for breeding stress-tolerant crop varieties.

## 2. Comparative Analysis of Physiological Responses

### 2.1. Individual Stress Responses

Drought stress typically results in reduced stomatal conductance, lower transpiration rates, and decreased photosynthetic activity, as plants close their stomata to conserve water [1,8,14]. Heat stress is characterized by elevated temperatures that destabilize cellular membranes, denature proteins, and impair photosynthetic processes [15,16]. Stomatal behavior under heat stress is complex. Initially, stomata may open to facilitate leaf cooling through transpiration. However, as heat stress persists, stomatal closure occurs to prevent excessive water loss. In soybean (Glycine max), stomatal conductance (gs) showed an initial increase under moderate heat stress but declined sharply as temperature exceeded 40 °C, indicating a shift from a cooling to a water conservation strategy [17].

In addition to stomatal regulation, drought stress typically results in declines in relative water content (RWC) and leaf turgor loss, leading to visible wilting. In strawberry (*Fragaria × ananassa*), the RWC dropped significantly under water-deficit conditions, particularly in non-inoculated plants, indicating an impaired water retention capacity [18]. These physiological adjustments reflect a trade-off between water conservation and carbon gain, often resulting in growth inhibition and early senescence under prolonged drought stress.

Heat stress primarily impairs photosynthesis and disrupts cellular homeostasis by increasing leaf temperatures and damaging the photosynthetic apparatus [2,19]. Elevated temperatures disrupt the stability of photosystem II (PSII), inhibit Rubisco activity, and reduce carbon assimilation rates [20]. In *P. yunnanensis*, exposure to 40 °C led to a significant reduction in chlorophyll content and increased leaf wilting, indicating thermal damage to the photosynthetic apparatus [21].

In *C. sinensis*, heat stress (42 °C) caused leaf curling and necrosis within 6 h, with damage severity increasing over time. Chlorophyll fluorescence imaging revealed a progressive decline in PSII efficiency, correlating with increased oxidative damage [22]. Moreover, heat stress significantly impairs pollen viability and fruit set. In tomato (Solanum lycopersicum), daytime temperatures above 30 °C and nighttime temperatures above 21 °C drastically reduced pollen germination and pollen tube elongation, leading to flower abortion [23].

### 2.2. Combined Stress Responses

Differently, under combined drought + heat stress, plants experience simultaneous water deficits and elevated temperatures, resulting in a complex interplay of physiological responses. This combined stress scenario is increasingly common under climate change and poses a significant threat to plant performance, particularly during reproductive stages.

#### 2.2.1. Photosynthetic and Stomatal Responses

One of the most consistent physiological responses to combined drought and heat stress is a rapid and sustained decline in the photosynthetic rate (Pn). Stomatal conductance (gs) also shows a more pronounced and persistent reduction under combined stress. In soybean, while heat stress alone caused transient fluctuations in gs, the addition of drought stress led to a continuous decline, with gs dropping by over 60% by day 6 of treatment [17]. This sustained closure limits water loss but also restricts CO_2_ uptake, exacerbating photosynthetic inhibition. The counteracting effects on stomatal regulation—where drought induces closure and heat promotes opening—lead to non-linear changes in photosynthetic rates and transpiration. Additionally, the integration of multiple signal transduction pathways (ABA, MAPK, CDPK, and ROS) results in a more pronounced impairment of the photosynthetic system and overall water-use efficiency [5,6,24,25]. Recent experimental work in barley and soybean supports these interactive effects, demonstrating that combined stress can exacerbate physiological dysfunction beyond the effects of each stress alone [10,12]. In Camellia sinensis, combined stress (42 °C + 20% PEG-6000) led to a 57.26% reduction in Pn after 24 h, significantly greater than that of either drought (11.61%) or heat (32.07%) stress alone [22]. This decline is attributed to both stomatal limitation (reduced CO_2_ diffusion) and non-stomatal factors, including thylakoid membrane damage and Rubisco inhibition [20].

#### 2.2.2. Water Status and Leaf Damage

Combined stress significantly accelerates leaf water loss and tissue damage. In *C. sinensis*, leaf curling and wilting were observed as early as 3 h under combined stress, compared to 6–12 h under single stresses [22]. The relative water content (RWC) also declined more sharply. In strawberry (*Fragaria* × *ananassa*), the RWC dropped to ~50% under combined drought and heat stress (40 °C day/12 °C night), whereas inoculation with Antarctic endophytic fungi (*Penicillium* spp.) helped maintain it near control levels [18].

Moreover, chlorophyll degradation and membrane injury were more severe under combined stress. Chlorophyll fluorescence imaging in tea plants revealed extensive PSII damage after 12 h of combined stress, with blue-green fluorescence indicating irreversible tissue injury [22]. This is consistent with the observed increase in malondialdehyde (MDA) content, a lipid peroxidation marker, which rose by 34.02% under combined stress, compared to 7.22% (drought) and 20.10% (heat) under single-stress conditions [22].

#### 2.2.3. Reproductive Impacts

Combined drought and heat stress has disproportionate effects on reproductive development. In maize, simultaneous exposure to water deficit and elevated temperatures during pollination led to kernel abortion and reduced grain set, even when vegetative tissues remained relatively unaffected [26,27]. This is due to the high sensitivity of floral organs to carbon shortage and oxidative stress under combined stress.

In tomato, combined stress (35/27 °C day/night + water deficit) resulted in pollen sterility and fruit abortion, with CWIN activity in anthers dropping below detectable levels [23]. Interestingly, emasculation and pollination with non-stressed pollen reduced abortion rates, indicating that male gametophyte development is a primary target of combined stress [27].

#### 2.2.4. Whole-Plant Water-Use Efficiency

Despite the decline in photosynthesis, water-use efficiency (WUE)—defined as the ratio of carbon gained to water lost—can sometimes improve under combined stress due to disproportionate reductions in transpiration. In strawberry, WUE was significantly higher in drought- and heat-stressed plants colonized by endophytic fungi, suggesting that microbial symbionts can enhance WUE by modulating stomatal behavior and hydraulic conductance [18].

However, this improvement in WUE often comes at the cost of biomass accumulation and yield. In *P. yunnanensis*, seedlings treated with GA (which promotes growth but reduces defense) showed severe wilting and reduced survival under combined stress, whereas JA-treated plants maintained a higher RWC and photosynthetic capacity [21]. This highlights the growth–defense trade-off that becomes particularly critical under combined stress conditions.

## 3. Comparative Analysis of Biochemical Responses

### 3.1. Individual Stress Responses

Plants reprogram primary and secondary metabolism within minutes to hours after the onset of a single stress. Recent metabolomics and enzymatic studies—mostly from the six source papers—reveal that drought and heat impose both unique and shared biochemical signatures that precondition later molecular events.

#### 3.1.1. Osmolyte and Hormonal Reprogramming

Proline remains the most universal osmolyte under drought stress. Drought stress triggers the accumulation of osmolytes such as proline and soluble sugars to maintain cellular osmotic balance [8,14,28]. In Camellia sinensis, proline increased by 7% after 24 h of 20% PEG-6000, while heat alone caused a 90% rise; the two stresses combined pushed the value to 6-fold [22], corroborating earlier maize ovary data, where proline began to accumulate at −0.8 MPa leaf water potential and reached 40 µmol g^−1^ DW just before abortion [29]. The biosynthetic step is catalyzed by Δ^1^-pyrroline-5-carboxylate synthetase (P5CS), whose gene (CsP5CR) is transcriptionally activated within 3 h of drought stress but repressed after 6 h of heat stress, indicating isoform-specific control [30].

γ-Aminobutyric acid (GABA) behaves oppositely: heat (42 °C) triggered a 4.4-fold rise in *C. sinensis* leaves within 24 h, whereas drought alone caused only a 0.36-fold rise [22]. Silencing CsGAD1 abolished GABA accumulation and simultaneously increased MDA by 78%, directly linking GABA shunt activity to membrane thermostability [31].

Abscisic acid (ABA) is drought-specific. In soybean, leaf ABA remained stable at 30 °C but rose linearly from 48 to 192 ng g^−1^ FW once irrigation was withheld, reaching significance 48 h before any decline in stomatal conductance [17]; this is fully consistent with the model that root-sourced ABA is transported xylem-ward to initiate stomatal closure [32,33].

Jasmonic acid (JA) is temperature-sensitive. In soybean leaves maintained at 40 °C, JA concentration doubled within 6 h, while ABA remained unchanged; ibuprofen-mediated inhibition of JA biosynthesis simultaneously suppressed small-HSP expression and increased electrolyte leakage, confirming JA as a thermo-sensory hormone [17,34].

#### 3.1.2. Soluble Sugar and Invertase Networks

Drought favors sucrose accumulation via the activation of sucrose-phosphate synthase (SPS) and the repression of acid invertase (AI). In *Pinus yunnanensis* JA-treated seedlings, the soluble sugar content remained stable during 96 h of water withholding, whereas in GA-treated plants, it decreased by 30% [21], mirroring earlier rice data where drought-tolerant genotypes maintained higher SPS/AI activity ratios [35].

Heat, in contrast, enhanced vacuolar invertase (VIN) activity and hexose pooling. Maturing tomato pollen exposed to 35 °C exhibited a 2-fold rise in VIN activity and a concomitant 40% increase in glucose + fructose, contributing to osmotic adjustment and protein stabilization [23]. Conversely, cell wall invertase (CWIN) activity declined under heat due to the induction of the inhibitor INVINH1 [36], illustrating isoform-specific regulation of invertases by temperature.

#### 3.1.3. Antioxidant and Oxidative Metabolism

Drought stress rapidly activates catalase (CAT) and peroxidase (POD). In JA-treated *P. yunnanensis* needles, 1.753 U g^−1^ FW CAT and 1.204 U g^−1^ FW POD were observed at 24 h, values 5- and 4-fold higher than those of GA-treated plants, and MDA was simultaneously suppressed by 22% [21]. Similar JA-primed antioxidant bursts have been reported in Brassica rapa, where the CAT2 transcript increased 3-fold, and H_2_O_2_ declined by 35% [37].

Drought stress is usually accompanied by a burst of reactive oxygen species (ROS), forcing plants to rapidly produce antioxidant enzymes, such as SOD, CAT, and POD, in order to contain oxidative damage [38,39]. High-temperature stress, in contrast, first triggers the synthesis of heat shock proteins (HSPs) to prevent protein denaturation and aggregation [19,40], yet it also generates abundant ROS that must be detoxified by the same antioxidant machinery [10,41]. Within minutes, both stresses activate Ca^2+^ signatures and MAPK cascades that act as emergency switches, integrating and amplifying downstream tolerance responses [18]. Core signaling hubs—including ABA and MAPK pathways—then cross-regulate antioxidant defenses, protein protection, and osmotic adjustment to ensure that the biochemical program is executed in a precise and orderly manner.

Heat stress elevates anthocyanin and flavonoid pools as UV/heat shields. *C. sinensis* leaves accumulated 2-fold more total anthocyanins after 24 h at 42 °C, coinciding with the upregulation of CHS and DFR genes. MDA levels still rose by 20% under heat stress, but the co-application of 1 mM GABA lowered MDA by 24%, revealing a complementary antioxidant role for GABA [22].

Heat stress induces the synthesis of HSPs to protect against protein denaturation and aggregation [19,40]. Concurrently, heat-induced ROS production requires robust antioxidant defenses to prevent cellular damage [10,38,41]. Calcium signaling and MAPK cascades are rapidly activated under heat stress, coordinating the biochemical responses necessary for thermotolerance [42]. At the protein level, heat causes the partial unfolding of nascent polypeptides and the aggregation of mature enzymes, most notably the oxygen-evolving complex (OEC) of photosystem II (PSII). Chloroplast stromal HSP70 and HSP90 are rapidly titrated away from their housekeeping roles to act as holdases, preventing irreversible aggregation [43,44,45]. Within 5 min, the nuclear master regulator HsfA1a trimerizes and binds to heat shock elements (HSEs), activating a transcriptional wave that includes HsfA2, HsfA3, and Hsp18.2 [46]. Cryo-EM structures reveal that the SUMOylation of HsfA1a stabilizes its trimeric form and increases DNA-binding affinity 3-fold during prolonged heat exposure [47,48,49]. The production of HSPs, driven by heat shock signaling via HSFs, is critical for protein refolding. Moreover, calcium influx and ROS serve as key secondary messengers that activate MAPK cascades, ultimately regulating stomatal behavior and water-use efficiency [50,51].

### 3.2. Combined Stress Responses: Metabolite Accumulation and Hormonal Crosstalk

In response to combined heat and drought (DH) stress, plants undergo profound biochemical reprogramming. These changes are not merely additive effects of individual stresses but represent unique metabolic signatures, including the accumulation of specific osmolytes, secondary metabolites, and phytohormones that collectively enhance stress tolerance.

#### 3.2.1. Proline and Polyamine Metabolism

The accumulation of proline is a hallmark of plant stress responses, functioning as an osmolyte, ROS scavenger, and molecular chaperone. Under DH conditions, proline biosynthesis is upregulated, while its degradation is suppressed. In tea plants, proline content increased by 6.08-fold under HS-DS, significantly higher than under single stresses. This accumulation is regulated by key genes such as CsP5CR and CsP5CDH, which are involved in proline biosynthesis and catabolism, respectively. Notably, CsP5CDH expression was downregulated under HS-DS, suggesting reduced proline turnover and enhanced accumulation [22].

Proline metabolism is intricately linked with polyamine biosynthesis, both of which share common precursors such as ornithine and arginine. Under HS-DS, polyamines such as spermidine and spermine also accumulate, contributing to membrane stabilization and ROS detoxification [52]. The interplay between proline and polyamine pathways under HS-DS suggests a coordinated metabolic response that supports cellular integrity and stress adaptation.

#### 3.2.2. Hormonal Regulation: ABA and JA

Phytohormones are central regulators of stress responses. ABA is a key hormone in drought signaling, promoting stomatal closure and osmotic adjustment. Under HS-DS, ABA levels increase significantly, facilitating stress adaptation [53,54]. ABA signaling, which is pivotal under drought stress, also modulates antioxidant responses when heat stress is present, creating a unique crosstalk with ROS and calcium signaling pathways [55,56]. Non-linear changes in ROS levels and antioxidant enzyme activities have been observed, indicating the presence of sophisticated regulatory mechanisms that enable plants to cope with dual stresses [57]. Recent research in tomato and barley has demonstrated distinct metabolite accumulation patterns that are unique to combined stress conditions [25].

However, ABA alone is insufficient to explain the complexity of DH responses, especially under high-temperature conditions, where jasmonates (JAs) become increasingly important. Jasmonic acid (JA) and its derivatives are lipid-derived hormones that regulate both biotic and abiotic stress responses. Under HS-DS, JA levels rise, particularly in heat-sensitive tissues, such as pollen and developing fruits [58]. JA signaling is mediated by the COI1-JAZ-MYC2 module, where JAZ proteins repress MYC2 transcription factors under non-stress conditions. Under stress, JA promotes the degradation of JAZ proteins, releasing MYC2 to activate downstream stress-responsive genes [59,60].

In Pinus yunnanensis, the co-application of methyl jasmonate (MeJA) and gibberellin (GA) modulated the expression of DELLA, JAZ, and MYC2 genes, indicating a hormonal crosstalk that balances growth and defense under HS-DS [21]. MeJA treatment enhanced antioxidant enzyme activities (CAT and POD), reduced wilting, and maintained the chlorophyll content, suggesting that JA plays a protective role under compound stress. The antagonistic interaction between GA and JA further supports the notion of a growth–defense trade-off, where JA prioritizes stress tolerance at the expense of growth [61,62].

#### 3.2.3. GABA and TCA Cycle Reconfiguration

GABA is synthesized primarily via the decarboxylation of glutamate by glutamate decarboxylase (GAD), and its accumulation is closely linked to the tricarboxylic acid (TCA) cycle. Under HS-DS, the TCA cycle is often downregulated, leading to reduced energy production. GABA serves as a metabolic bypass, entering the TCA cycle via the GABA shunt, which produces succinate and helps maintain mitochondrial function under stress [31]. Furthermore, the exogenous application of GABA has been shown to reduce oxidative damage, decrease malondialdehyde (MDA) and hydrogen peroxide (H_2_O_2_) levels, and enhance antioxidant enzyme activities [20], indicating its protective role in cellular homeostasis.

#### 3.2.4. Integration of Metabolic and Hormonal Pathways

The biochemical responses to DH stress are not isolated events but represent an integrated network of metabolic and hormonal signaling. For instance, GABA accumulation is not only a metabolic response but also influences ABA and JA signaling pathways [63]. Similarly, proline and polyamines modulate ROS levels, which, in turn, affect hormone biosynthesis and signaling [30,52].

Moreover, sugar metabolism and hormonal signaling are tightly interlinked. Invertases (CWIN, VIN, and CIN) play dual roles in sugar metabolism and signaling, influencing hormone biosynthesis and responsiveness [27]. Under HS-DS, invertase activity is often suppressed, leading to reduced hexose availability and altered sugar signaling, which further impacts hormone balance and gene expression [26,64].

## 4. Comparative Analysis of Molecular Responses

At the molecular level, plants respond to combined heat and drought stress (HS-DS) through the extensive reprogramming of gene expression networks, the activation of stress-specific signaling pathways, and the modulation of hormonal crosstalk. These responses are highly coordinated and involve transcription factors (TFs), hormone-responsive genes, and metabolic enzymes that collectively determine the plant’s ability to survive and adapt under compound stress conditions. This section focuses on the key molecular players and regulatory modules involved in HS-DS responses, with emphasis on the roles of abscisic acid (ABA), jasmonates (JAs), gibberellins (GAs), and their downstream signaling components.

### 4.1. Individual Stress Responses

#### 4.1.1. Ca^2+^-Mediated Membrane-to-Nucleus Signaling

Within seconds of a heat upshift, lipid microdomains open Ca^2+^-permeable channels [8]. The transient cytosolic Ca^2+^ spike is decoded by calmodulins (CaMs) and calcium-dependent protein kinases (CPKs), which phosphorylate HSFA1a and promote its trimerization [65]. In *P. yunnanensis*, the PyCPK3 transcript peaks at 15 min under 40 °C but remains unchanged under PEG-induced drought stress, confirming heat specificity [21]. Conversely, osmotic stress triggers Ca^2+^ release from internal stores via mechanosensitive OSCA channels [66]; the resulting Ca^2+^ signature is longer lasting (>30 min) and preferentially activates CPK4/11–AREB1 phosphorylation cascades that are absent in pure-heat treatments [42].

#### 4.1.2. Reactive Oxygen Species (ROS) Wave and MAPK Phosphorylation

Both stresses elevate apoplastic ROS but with different kinetic signatures. Heat produces a rapid H_2_O_2_ burst (<5 min) via NADPH oxidase RBOHD, which is essential for HSFA1a nuclear translocation [67]. rbohD mutants fail to induce HSP70 under 38 °C yet still respond normally to dehydration, illustrating that the ROS wave is dispensable for drought signaling [48]. Instead, water deficit activates MPK3/MPK6 modules that phosphorylate AREB1 on Ser-43, a residue not targeted during heat stress [68]. Consistent with this, *P. yunnanensis* shows strong MPK6 activation (the p44/p42 band intensity increases 3.2-fold) after 2 h of drought stress but a negligible change under heat stress [21].

#### 4.1.3. Hormone Biosynthesis Genes: Differential Rate-Limiting Steps

ABA biosynthesis is drought-exclusive. NCED3 expression rises >50-fold within 30 min of the soil water potential dropping to −0.6 MPa but is undetectable under 40 °C [69]. Conversely, heat rapidly induces LOX3 and AOS (jasmonate biosynthesis), producing a JA-Ile peak at 1 h that returns to baseline by 3 h [42]. Gibberellin deactivation is also heat-specific: the GA2ox8 transcript doubles every 30 min up to 3 h at 40 °C, leading to DELLA accumulation and growth arrest [21]. Under drought stress, GA2ox genes remain unchanged, while GA20ox1 is actually upregulated in elongating roots, indicating that GA catabolism is not a universal stress response but is selectively recruited by thermal stress.

#### 4.1.4. Metabolic Gene Modules: Proline vs. GABA Shunt

Proline biosynthesis (P5CS1) is exclusively drought-induced; its promoter contains two drought-responsive ABRE motifs that are not recognized by heat-activated TFs [30]. GABA shunt genes (GAD1 and GABA-T) remain unaltered under drought but are strongly upregulated by heat, supplying succinate to maintain TCA cycle flux when aconitase activity is thermolabile [31]. Thus, the choice of osmolyte—proline under drought stress and GABA under heat stress—reflects distinct metabolic constraints rather than quantitative differences in stress intensity.

### 4.2. Combined Stress Responses: Decoding Signaling Hubs and Transcriptional Trade-Offs

#### 4.2.1. Hormone Signaling Networks: From Antagonistic Circuits to Convergent Nodes

Combined heat–drought (H+D) stress is perceived by plants as a non-additive signal that rewires hormone circuitry far beyond the simple superposition of single-stress responses [41]. ABA is rapidly accumulated within 30 min of leaf dehydration and acts as the primary systemic alarm signal [26]. Under H+D stress, guard-cell ABA levels increase 4-fold compared with those under drought stress alone, a synergism that is synchronized by a heat-accelerated xylem sap flow [70]. The canonical PYR/PYL–PP2C–SnRK2 kinase cascade is fully activated, phosphorylating ABF/AREB TFs that occupy ABRE cis-elements in promoters of late-embryogenesis-abundant (LEA) genes, dehydrins (DHNs), and antioxidant enzymes [71]. However, parallel heat-induced accumulation of JA-Ile (about 2.3-fold within 1 h) introduces a negative crosstalk node: MYC2, the master JA TF, competes with AREB2 for the same ABRE-like motifs, thereby attenuating ABA-driven stomatal closure while potentiating ROS scavenging [61,72]. The DELLA–JAZ co-repressor module acts as a molecular rheostat: DELLA proteins physically sequester JAZs, releasing MYC2 and simultaneously repressing GA responses that would otherwise promote growth at the expense of defense [62,73]. Transcriptomic time courses in soybean revealed that this DELLA–JAZ balance is tipped within 2 h of H+D stress onset, preceding any photosynthetic decline [17]. GA catabolism is reinforced by heat-inducible GA2ox genes, whose promoters harbor both ABRE and TGACG (MeJA-responsive) elements, providing a cis-integrative platform for the two hormones [74]. Thus, the early H+D signaling network is best viewed as a competitive TF marketplace where ABA, JA, and GA signals are simultaneously translated into a coherent transcriptional output that prioritizes stress tolerance over growth.

#### 4.2.2. Sugar Sensing and Invertase-Mediated Metabolic Checkpoints

Sugar status is no longer regarded as a passive downstream indicator but rather as an active determinant of H+D acclimation [75]. Within 1 d of stress, phloem sucrose import into maize ovaries drops by 70%, triggering a switch from sink to source limitation [76]. Cell wall invertase (CWIN) activity at the phloem termini collapses concomitantly, not because of transcript downregulation but via post-translational interaction with INVINH1, whose promoter is synergistically activated by ABA and heat-induced ROS [36,64]. Consequently, proline accumulates to 60 µmol g^−1^ FW, supplying both osmotic adjustment and NADP+ for the chloroplastic malate valve [30]. Vacuolar invertase (VIN) exhibits an opposite kinetic pattern: its activity is transiently elevated 2-fold in heat-tolerant tomato lines, facilitating hexose build-up that drives cell expansion and maintains pollen turgor [23]. Cytosolic invertase (CIN), long considered to perform housekeeping functions, emerges as a specific H+D node; its loss of function in Arabidopsis (cinv1; cinv2) exaggerates ROS bursts and blocks the induction of APX2, leading to pollen sterility [77]. Functional reversion experiments—the stem infusion of ^13^C-sucrose during H+D stress—demonstrate that the restoration of CWIN and VIN activities recovers 68% of grain set, pinpointing invertases as metabolic gatekeepers whose activity integrates hormonal and sugar cues into reproductive fate [78].

#### 4.2.3. Heat Shock Factors and ROS-Dependent Transcriptional Hubs

Heat shock factors (HSFs) constitute the third regulatory layer. A genome-wide analysis in wheat identified 81 TaHSFs, of which clade A1 members (TaHSFA2-7B and TaHSFA6-D) were upregulated >200-fold within 30 min of H+D stress, preceding the expression of canonical HSP70/90 chaperones [79]. Promoter dissection revealed an H+D-specific cis-module composed of three heat shock elements (HSEs) flanked by two ABREs; the deletion of either HSE or ABRE abolished reporter activity, indicating that HSFs act as stress-specific integrators rather than simple thermometers [80]. Interestingly, the same clade-A1 HSFs trans-activate the promoters of cytosolic APX1 and CAT2 by binding to a non-canonical HSE (nGAAn)3, thereby coupling protein quality control with ROS detoxification [81]. ROS themselves feed back into the network: H_2_O_2_ at 5 mM triggers the nuclear export of HSFAs within 15 min, dampening the heat response once redox homeostasis is re-established [77]. This ROS-HSF feedback loop is fine-tuned by aquaporin-facilitated H_2_O_2_ translocation; the silencing of FaPIP2;1 in strawberry restricts H_2_O_2_ efflux from chloroplasts, prolonging HSF nuclear retention and enhancing thermotolerance [18]. Thus, H+D acclimation relies on a self-limiting HSF–ROS oscillator that aligns chaperone capacity with cellular redox demand.

#### 4.2.4. Epigenetic Priming and Chromatin Memory

Recent evidence places chromatin remodeling as a fourth mechanistic stratum. Three-day priming with mild heat (35 °C) followed by drought imprints an open chromatin state at the PYL5 and DREB2B loci that persists for at least 5 d, facilitated by the eviction of histone H2A.Z and the enrichment of H3K4me3 [82]. This epigenetic memory requires the histone chaperone ASF1, whose transcript is itself HSF-dependent, thereby linking heat sensing to chromatin accessibility [83]. JA signaling intersects via IBM1, a JmjC-domain demethylase that removes H3K9me2 from JA-responsive genes; IBM1 mutants fail to prime DREB2B and consequently collapse under repeated H+D cycles [84]. Importantly, DNA methylation exhibits locus-specific plasticity: CG hypomethylation within the promoter of VIN1 is associated with sustained transcription during recovery, whereas CHH hypermethylation of CWIN3 silences its expression, favoring hexose accumulation in primed seedlings [51]. Collectively, these data portray H+D priming as a multi-layered chromatin relay that converts transient hormonal and metabolic inputs into durable transcriptional competence.

## 5. Future Perspectives

### 5.1. From Linear Pathways to Resilient Networks: Exploiting WGCNA and Machine Learning-Guided Systems Genetics

Weighted gene co-expression network analysis (WGCNA) has matured from a descriptive tool into a predictive engine. By integrating physiological phenotypes (e.g., Δgs, A/Ci curvature, and quantum efficiency) with time-resolved transcriptomes, WGCNA can now identify modules that explain >70% of phenotypic variance under combined heat–drought (H+D) scenarios [25]. Recent barley studies revealed a cyanogenic β-glucosidase hub gene that, when edited, simultaneously stabilizes stomatal kinetics and maintains spikelet fertility under H+D stress [85]. Extending this strategy to perennial species requires embedding circadian and seasonal covariates into network inference; random forest-assisted WGCNA (RF-WGCNA) pipelines now allow >50 environmental vectors to be weighted alongside gene expression, markedly increasing the predictive power of hub gene discovery [86].

### 5.2. Toward a Unified Multi-Omics Space: From Correlation to Causality

The next frontier is not simply layering omics layers but rather constraining them by enzyme kinetics and thermodynamics. Recent rice work demonstrated that coupling the absolute quantification of ABA and JA-Ile to phospho-proteomics enables the construction of mass-action models that accurately forecast MAPK activation thresholds under H+D stress [87]. Cloud-based platforms such as PlantCube now integrate transcriptomic, phospho-proteomic, and primary metabolite data into kinetic models that predict carbon allocation errors before phenotypic collapse [88]. Importantly, the inclusion of single-cell or spatial metabolomics is revealing cell-type-specific invertase–sugar hubs that are invisible in bulk samples, offering entry points for cell-targeted genome editing [84].

### 5.3. Genome Editing 2.0: Multiplexed, Precise, and Cis-Regulatory

Early transgenic stacks (e.g., AVP1 + SIZ1 or AtHXK1 + SP6A) validated the additive value of combining drought- and heat-specific genes yet suffered from linkage drag and consumer resistance [89]. CRISPR-Cas derivatives now enable multiplexed promoter fine-tuning without foreign DNA. For example, the removal of a 21 bp ABRE–HSE composite repressor from the GA2ox8 promoter increased bioactive GA locally in maize ovaries, accelerated silking, and shortened the anthesis-to-silking interval by 1.8 d under H+D stress, translating into a 12% yield advantage in field trials [21]. Base-editing of the −45 bp position of the TaCWIN1 promoter created a de novo ABRE motif, boosting kernel CWIN activity 1.7-fold and raising the thousand-kernel weight under H+D stress without extra irrigation [27]. Such cis-regulatory edits circumvent the yield penalties observed with constitutive over-expression and are likely to face lower regulatory hurdles.

### 5.4. Breeding by Design: Integrating Genomic Selection with Mechanistic Priors

Genomic selection (GS) has already delivered commercial drought-tolerant maize hybrids [90], but prediction accuracy for combined stress remains modest due to G × E × M interactions. Embedding mechanistic priors—e.g., allelic effects on ABA sensitivity or invertase kinetics—into Bayesian GS models improved prediction accuracy by 18% for H+D tolerance in a recent 384-line rice diversity panel [91]. Looking forward, coupling real-time phenomics from UAV or low-orbit sensors with GS will allow for environment-specific genomic estimates, enabling dynamic selection trajectories that match seasonal climate forecasts [92].

### 5.5. Current Challenges in Stress Response Research

Despite the significant progress made, several challenges remain. Standardizing experimental conditions and integrating diverse datasets across studies continue to be major hurdles. Moreover, the interactive mechanisms under combined stress conditions are only partially understood. Recent reviews have highlighted the need for more robust, multi-scale experiments to fully capture the dynamic nature of stress responses [5,6,39]. The integration of detailed signal transduction studies into these experiments is critical for a holistic understanding. Future research should emphasize long-term, multi-scale experiments that combine advanced transcriptomic, proteomic, and metabolomic approaches with systems biology and big data analytics [88,93]. Emphasis on elucidating the crosstalk between various signaling pathways—such as ABA, MAPK, and calcium-mediated networks—will be essential. Interdisciplinary and international collaborations will be crucial for unraveling these complex regulatory networks and identifying novel molecular targets for breeding. Recent advancements in gene editing and multi-omics integration offer promising avenues to further enhance our understanding and the application of stress tolerance mechanisms [25,84,85].

## 6. Conclusions

Drought and heat stress, when experienced together, represent a distinct and uniquely challenging stress paradigm—not merely an additive combination of two individual stresses. Plants perceive this compound stress as a novel environmental signal, triggering a non-linear, highly integrated response that rewires hormonal signaling, reprograms metabolism, and reshapes gene expression in ways that single stresses do not.

At the core of this response is a competitive hormonal signaling network, where abscisic acid (ABA), jasmonic acid (JA), and gibberellin (GA) converge on shared transcriptional hubs. These hormones do not act in isolation; instead, they engage in cross-regulatory antagonism and synergy, fine-tuning stomatal behavior, sugar metabolism, and reproductive development. This hormonal crosstalk is further modulated by metabolic checkpoints such as invertase–sugar modules and ROS-HSF oscillators, which act as molecular gatekeepers, determining whether the plant adapts, survives, or succumbs.

Importantly, these responses are not hardwired—they are context-dependent and dynamically regulated, influenced by the timing, intensity, and duration of stress. This complexity, while challenging, also presents unprecedented opportunities for crop improvement. By leveraging systems-level tools—such as weighted gene co-expression networks, kinetic multi-omics models, and cis-regulatory CRISPR editing—we can now precisely engineer stress resilience without compromising yield.

## Figures and Tables

**Figure 1 cimb-47-00975-f001:**
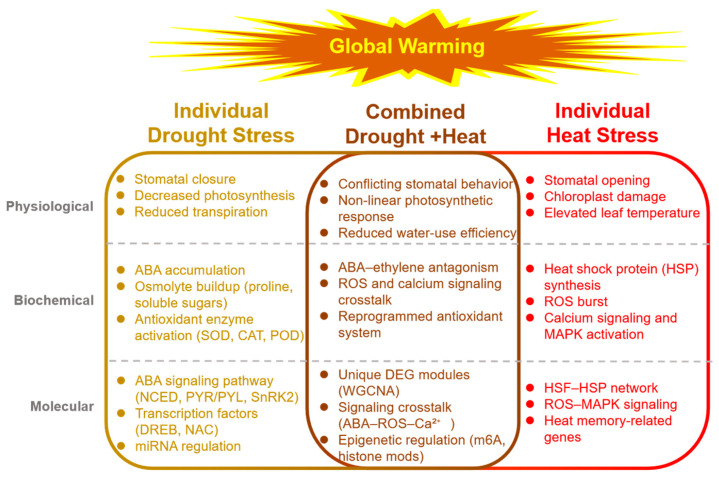
Integrated overview of plant responses to individual and combined drought and heat stress. Note: (**Left**) Drought stress triggers stomatal closure, ABA accumulation, osmolyte synthesis, and antioxidant enzyme activation; key molecular players include NCED, PYR/PYL, SnRK2, DREB, and NAC TFs. (**Right**) Heat stress induces stomatal opening, HSP synthesis, ROS burst, and calcium-MAPK signaling; master regulators comprise HSFs, HSPs, and ROS-responsive TFs. (**Center**) Combined drought + heat stress generates conflicting stomatal signals, non-linear photosynthetic responses, and unique biochemical and transcriptional signatures characterized by ABA–ethylene antagonism, ROS–calcium crosstalk, and stress-specific gene modules identified by multi-omics analyses. Arrows indicate positive or regulatory interactions; barred lines denote inhibitory effects. The central circle highlights the core signaling hubs (ABA, MAPK, Ca^2+^, ROS) that integrate and coordinate systemic stress adaptation.

**Figure 2 cimb-47-00975-f002:**
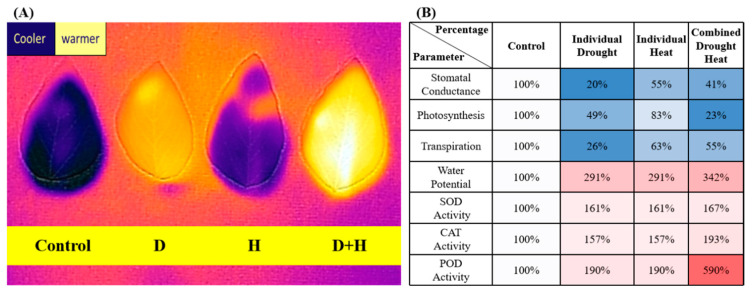
(**A**) Leaf temperature of soybean plants subjected to control, H (heat), D (drought), and combined drought and heat (D+H) stress conditions. The leaf temperature was measured using a “Flir-One” thermal imaging system (Flir, Nashua, NH, USA). (**B**) Measurement of physiological parameters in plants subjected to control, heat (H), drought (D), and drought and heat (DH) stress conditions. Reprinted from ref. [9].

## Data Availability

This article is a review and does not contain any original data. All references and sources cited in this review are publicly available and can be accessed through the provided citations. For any additional information or inquiries, please contact the corresponding author.

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
