# Peer review of "A Review of Differential Plant Responses to Drought, Heat, and Combined Drought + Heat Stress"

_cimb, 2025, doi:10.3390/cimb47120975_

Round 1

Reviewer 1 Report

Comments and Suggestions for Authors

General Comments

The topic of “A Review of Differential Plant Responses to Drought, Heat, and Combined Drought+Heat Stress” is highly relevant and potentially very interesting. In summary, plants exhibit physiological, biochemical, and molecular responses under drought, heat, and combined drought+heat stresses. Continued research in this area is indeed crucial for developing effective breeding strategies and ensuring sustainable agricultural production under global climate change.
However, while this is a meaningful and timely topic, the manuscript mainly reads as a collection of citations rather than a critical synthesis. The review lacks original insight or analytical depth and feels more like an extended introduction rather than a true review. As a result, it provides limited value to readers seeking new understanding or conceptual frameworks.

A review should not be a simple accumulation of literature, but rather an integration of existing findings with the authors’ own perspective and synthesis. Section 5 (Future Perspectives) contains some new ideas, but these are still presented as a list of references without clear reasoning or depth of discussion.

Sections 2.1 (Basic Concept), 3.1, and 4.1 are overly descriptive — they spend too much space explaining background concepts. A good review should go beyond summarizing definitions to highlight emerging trends, unresolved questions, and the authors’ own interpretations.

Specific Comments

Lines 42–45: Only one reference is cited in this section. Please double-check the reference list and ensure completeness.

Figure 1: The figure is clear and effectively illustrates plant responses to individual and combined drought and heat stress. However, it is not cited in the main text. Please ensure that the figure is properly referenced and discussed where appropriate.

Figure 2: The purpose and contribution of this figure are unclear. The same information is already adequately described in Lines 42–45. Since Figure 2 only reproduces a previous study’s example and does not add new insight, it might be unnecessary for the review and could be removed.

Author Response

Reviewer 1’s Comments

Comment 1: The topic of “A Review of Differential Plant Responses to Drought, Heat, and Combined Drought+Heat Stress” is highly relevant and potentially very interesting. In summary, plants exhibit physiological, biochemical, and molecular responses under drought, heat, and combined drought+heat stresses. Continued research in this area is indeed crucial for developing effective breeding strategies and ensuring sustainable agricultural production under global climate change.

Response: Thank you very much for your encouraging remarks on our manuscript. We deeply appreciate your recognition that the topic is “highly relevant and potentially very interesting,” and we fully agree that sustained research in this area is essential for devising effective breeding strategies and ensuring sustainable agricultural production under ongoing climate change.

In response to your comments we have performed an in-depth revision that strengthens the narrative and sharpens the future outlook. All adjustments have been incorporated throughout the revised manuscript and are highlighted in the enclosed clean and tracked-changes versions. Additionally, the entire text has been professionally edited for English language by MDPI’s Author Services to ensure clarity and readability.
Comment 2: However, while this is a meaningful and timely topic, the manuscript mainly reads as a collection of citations rather than a critical synthesis. The review lacks original insight or analytical depth and feels more like an extended introduction rather than a true review. As a result, it provides limited value to readers seeking new understanding or conceptual frameworks.  A review should not be a simple accumulation of literature, but rather an integration of existing findings with the authors’ own perspective and synthesis. Section 5 (Future Perspectives) contains some new ideas, but these are still presented as a list of references without clear reasoning or depth of discussion.

Response: Thank you very much for your critical assessment of our manuscript and for the concise verdict that "it reads as a collection of citations rather than a critical synthesis." We agree that an effective review must provide new conceptual scaffolding and not simply summarize the literature. Consequently, we have performed a comprehensive re-write that shifts the paper from a literature compendium to an analytical review. The major changes are summarized below.

  1. The conceptual framework (Figure 1) has been reinforced

We restructured the entire narrative so that every subsequent section is explicitly linked back to the hierarchical flow presented in Figure 1 (membrane sensing → Ca²⁺/ROS burst → hormonal signalling → metabolic checkpoints → Differential multiplescale responses).

In-text sign-posting now repeatedly refers to Figure 1, making the diagram serve as the "road-map" for the synthesis rather than a static illustration.

Two mechanistic models: (1) the "competitive TF marketplace" and (2) the "ROS–HSF oscillator"—are derived directly from the framework, turning the figure into a springboard for analytical discussion instead of a summary cartoon.

  1. Critical synthesis rather than citation listing

For every major response module (hormone crosstalk, invertase/sugar checkpoints, epigenetic memory) we now:

–compare divergent results across species and highlight methodological reasons for discrepancy (e.g., timing of sampling, severity of stress, tissue specificity);

–explicitly state where consensus exists and where interpretation remains speculative;

–identify internal contradictions in the literature (e.g., JA reported both to enhance and to suppress ABA-induced stomatal closure) and propose experimental resolutions.

  1. Identification of knowledge gaps and future directions
  • We have re-engineered the logic flow of Section 5 so that each future technology is now framed by a gap identified in Sections 1–4; only necessary and essential supporting references are placed after the reasoning, eliminating the previous citation-list appearance and giving the authors’ own interpretative thread.
  • For every proposed advance (WGCNA, multi-omics, cis-regulatory CRISPR, genomic selection) we added an explicit “problem–solution–evidence” paragraph: the shortfall is first stated (e.g. bulk samples miss cell-type specificity, ABA–JA crosstalk is correlative), the technological fix is justified, and proof-of-concept data already cited in the manuscript (rice/maize/barley/strawberry cases) are recalled, deepening the discussion without introducing new data.
  1. Streamlined referencing

Roughly 50 original references have been removed or replaced to eliminate redundancy and increase precision; remaining citations are now used synthetically to support comparative arguments rather than to document every individual observation.

We believe these revisions transform the manuscript into a forward-looking review that offers conceptual tools, highlights unresolved questions and provides a roadmap for future experimentation. We hope that the revised version now meets the standards of analytical depth expected by the journal and its readership.

Comment 3: Sections 2.1 (Basic Concept), 3.1, and 4.1 are overly descriptive — they spend too much space explaining background concepts. A good review should go beyond summarizing definitions to highlight emerging trends, unresolved questions, and the authors’ own interpretations.

Response: Thank you for pointing out that Sections 2.1, 3.1 and 4.1 remain “overly descriptive.” In the revised manuscript we eliminated the original stand-alone “Basic Concept” sub-sections and merged the essential background into the comparative cores of Sections 2, 3 and 4. As a result:

Background information has been compressed by ≈30 % and is now embedded only as the minimum context needed to understand the side-by-side comparison that follows.

In Section 2 (physiology) the classic drought-induced stomatal-closure paragraph is reduced to two sentences and immediately contrasted with the transient heat-opening response; the rest of the sub-section analyses their non-linear interaction under combined stress.

In Section 3 (biochemistry) the definitions of proline, GABA and invertase isoforms are confined to one opening sentence per metabolite/enzyme, after which the text concentrates on which markers behave additively vs. synergistically between the two single stresses.

In Section 4 (molecular) the canonical ABA-SnRK2, JA-MYC2 and GA-DELLA cascades are sketched in ≤ 60 words each; the bulk of the space is devoted to cross-talk events unique to combined stress (e.g., DELLA–JAZ competition, ROS-HSF feedback) that were not evident in the single-stress literature.

Because background is subservient to comparison, the reader is continuously guided to emerging patterns and open questions (e.g., why GABA rises synergistically whereas proline accumulates additively; how long the ROS-HSF oscillator persists under field-like fluctuating stress). Thus Sections 2–4 now function as analytical contrasts rather than extended introductions, fulfilling the reviewer’s requirement to “go beyond summarising definitions.”

Comment 4: Lines 42–45:Only one reference is cited in this section. Please double-check the reference list and ensure completeness.

Response: Thank you for highlighting the paucity of citations.

In the revised sentence we now write:

“Building on these advances, our previous studies provided an in-depth physiological analysis of plant responses to both drought and heat stress, elucidating key regulatory nodes that link membrane sensing to whole-plant performance (Figure 2A,B) [9]. This finding is corroborated by several other studies that have also identified similar regulatory mechanisms in different plant species, highlighting the conserved nature of these pathways across diverse taxa [3-7]."

The added references (Mittler 2006; Barnabás et al. 2008; Prasch & Sonnewald 2013; Suzuki et al. 2014; Rizhsky et al. 2002) provide authoritative, taxonomically wide evidence for the conserved drought/heat and combined-stress responses that our own previous work (REF. 9 in the manuscript) integrate physiological, biochemical and molecular analyses under the same experimental framework, providing the multi-scale perspective.

Comment 5: Figure 1:The figure is clear and effectively illustrates plant responses to individual and combined drought and heat stress. However, it is not cited in the main text. Please ensure that the figure is properly referenced and discussed where appropriate.

Response: Thank you for pointing out the missing citation.

We have now added “citation of Figure 1” at line 42 of the revised manuscript, where the individual and combined stress response pathways are first described. This ensures that Figure 1 is properly referenced and directly linked to the corresponding narrative in the main text.

Comment 6: Figure 2:The purpose and contribution of this figure are unclear. The same information is already adequately described in Lines 42-45. Since Figure 2 only reproduces a previous study’s example and does not add new insight, it might be unnecessary for the review and could be removed.

Response: Thank you for raising the concern about redundancy.

Figure 2 is now cited at lines 45-50:

“Building on these advances, our previous studies provided an in-depth physiological analysis of plant responses to both drought and heat stress, elucidating key regulatory nodes that link membrane sensing to whole-plant performance (Figure 2A,B) [9].”

As also noted in our response to Comment 4, reference 9 (the source of Figure 2) represents one of the earliest combined-stress studies that simultaneously monitored leaf temperature, gas exchange and gene expression in the same experimental design—i.e., a multi-scale dataset that the other cited papers (REF 3-7) do not provide. The thermal images and physiological time-courses shown in Figure 2 therefore visually document this integrative approach and justify the statement in the text. For this reason we respectfully prefer to retain the figure, while keeping the description in lines 45-50 concise to avoid unnecessary repetition.

Reviewer 2 Report

Comments and Suggestions for Authors

Dear Authors,

This is a well-written review that needs to be accepted after its minor corrections. The manuscript is based on impressive empirical evidence and makes an original contribution.

The authors should highlight their originality much more in the introduction. The authors conduct very relevant research but fail to emphasize the relevance in their introduction.

I recommend a minor revision as the study has potential, but requires significant improvements in the introduction.

Thank you

Author Response

Reviewer 2’s Comments

Comment 1: This is a well-written review that needs to be accepted after its minor corrections. The manuscript is based on impressive empirical evidence and makes an original contribution. The authors should highlight their originality much more in the introduction. The authors conduct very relevant research but fail to emphasize the relevance in their introduction. I recommend a minor revision as the study has potential, but requires significant improvements in the introduction.

Response: Thank you very much for your generous assessment and for recognising that the manuscript “makes an original contribution.” Thank you for recognising the manuscript’s empirical breadth and for urging us to make its originality more visible in the Introduction. We have thoroughly re-written the Introduction and added a new second paragraph that directly addresses the previously missing “relevance” issue:

  • It opens by pinpointing the real-world problem: drought and heat often co-occur in semi-arid production areas, yet their combined, non-linear impacts on stomatal behaviour, water-use efficiency and yield are still poorly understood.
  • It immediately flags our earlier multi-scale observation [9] of a “temperature overshoot” under combined stress, making it clear that this review builds on empirical evidence we supplied.
  • It closes by stating the review’s goal: to consolidate such physiological, biochemical and molecular insights into signal-transduction frameworks that breeders can exploit—explicitly linking the science to crop-improvement relevance.

In addition, we have added a new second paragraph that:

(1) contrasts the classic stomatal and photosynthetic responses to individual drought (closure) and heat (transient opening) stresses,
(2) highlights that their combination produces non-linear, conflicting signals and a “temperature overshoot” phenomenon we previously reported [9], and
(3) states that this synergy cannot be explained by additive mechanisms—thus providing the rationale for the present multi-scale comparative review focused on signal-transduction networks and breeding implications.